# A New NT4 Peptide-Based Drug Delivery System for Cancer Treatment

**DOI:** 10.3390/molecules25051088

**Published:** 2020-02-28

**Authors:** Jlenia Brunetti, Sara Piantini, Marco Fragai, Silvia Scali, Giulia Cipriani, Lorenzo Depau, Alessandro Pini, Chiara Falciani, Stefano Menichetti, Luisa Bracci

**Affiliations:** 1Department of Medical Biotechnology, University of Siena, via Aldo Moro 2, 53100 Siena, Italy; silvia.scali@unisi.it (S.S.); lorenzo.depau@unisi.it (L.D.); pinia@unisi.it (A.P.); chiara.falciani@unisi.it (C.F.); luisa.bracci@unisi.it (L.B.); 2Department of Chemistry “Ugo Schiff”, University of Florence, Via Della Lastruccia 3-13, 50019 Sesto Fiorentino (FI), Italy; sarapiantini85@gmail.com (S.P.); giuliacipr@gmail.com (G.C.); stefano.menichetti@unifi.it (S.M.); 3Magnetic Resonance Center (CERM), via Luigi Sacconi 6, 50019 Sesto Fiorentino (FI), Italy; fragai@cerm.unifi.it

**Keywords:** drug delivery, selective tumor-targeting agent, theranostic nanodevices, paclitaxel

## Abstract

The development of selective tumor targeting agents to deliver multiple units of chemotherapy drugs to cancer tissue would improve treatment efficacy and greatly advance progress in cancer therapy. Here we report a new drug delivery system based on a tetrabranched peptide known as NT4, which is a promising cancer theranostic by virtue of its high cancer selectivity. We developed NT4 directly conjugated with one, two, or three units of paclitaxel and an NT4-based nanosystem, using NIR-emitting quantum dots, loaded with the NT4 tumor-targeting agent and conjugated with paclitaxel, to obtain a NT4-QD-PTX nanodevice designed to simultaneously detect and kill tumor cells. The selective binding and in vitro cytotoxicity of NT4-QD-PTX were higher than for unlabeled QD-PTX when tested on the human colon adenocarcinoma cell line HT-29. NT4-QD-PTX tumor-targeted nanoparticles can be considered promising for early tumor detection and for the development of effective treatments combining simultaneous therapy and diagnosis.

## 1. Introduction

Drug delivery systems can enhance therapeutic effects and have fewer side- and adverse effects, as well as reduced chemotherapic toxicity. A challenge for pharmacology is to find the best formulation and the most suitable route of administration to achieve these goals. 

Peptide-based systems may be used in drug delivery, owing to specific chemical properties of peptides, which are easy to obtain by chemical synthesis and can be modified to deliver therapeutic agents including drugs and diagnostic molecules. These can be coupled to the carrier peptide by tailored chemical bonds [1,2,3].

Drawbacks related to peptides as drugs is their short half-life in vivo due to rapid protease degradation. In the last few years, several strategies have been considered to improve the metabolic stability through chemical approaches aimed to modify the original peptide sequences. A possible approach is the introduction of D or other unnatural amino acids, structural constraints, cyclization, or transformation of lead peptides into peptidomimetics or small molecules [4]. Peptide proteolytic cleavage can be dramatically reduced by synthetizing peptides in branched form, where multiple copies of the same peptide are linked covalently to a branched lysine core [5]. By allowing multimeric binding to peptide targets, synthesis in branched form can increase local peptide concentrations and biological activity. 

There are currently several peptide-based drug delivery nanosystems in clinical trial phase [6]. Nanoparticles, liposomes, and a variety of polymer-based materials are used for carrier construction. Nanoparticles are an extremely active field for new drug-delivering carrier development, with advantages over other systems. They improve the bioavailability and therapeutic efficiency of antitumor drugs, while accumulating preferentially at the tumor site [7,8,9].

Here we report the development of a peptide-based drug delivery system using the tetra-branched peptide NT4, which is a promising cancer theranostic by virtue of its high cancer selectivity. We previously reported a stable tetrabranched peptide, named NT4, which binds with high selectivity to human cancer cells and tissues and efficiently and selectively delivers drugs or tracers for cancer cell imaging or therapy. We found that the high selectivity of NT4 toward cancer cells and tissues resided in its high affinity binding to sulfated glycosaminoglycans [10,11,12,13,14].

In a breast cancer orthotopic mouse model, NT4 conjugated with paclitaxel caused tumor regression that was not achieved with paclitaxel alone under the same experimental conditions [11]. In fact, selective binding of NT4 to cancer cells enabled enhanced local concentrations of the drug in cancer tissues and increased tumor cytotoxicity. 

We recently constructed near-infrared quantum dots (QDs) functionalized with NT4 peptide (NT4-QDs). We observed specific uptake of NT4-QDs in human cancer cells in vitro, and much higher accumulation and retention of targeted QDs at the tumor site, compared to the non-targeted QDs, in a mouse colon cancer model [15]. To achieve higher tumor-accumulation of the drug, multiple drug moieties can be loaded on a single carrier, reducing the off-target toxicity of the free drug and maximizing treatment efficacy [16]. Functionalized nanosystems are precious tools as they can be turned into tumor specific probes and can be easily modulated in terms of scale of the carried cargo. The possibility of increasing or decreasing the molecular cargo of cytotoxic moieties is necessary to fine-tune selectivity and consequently decrease general toxicity. Based on our previous achievements [11,15], we tested different technical approaches aimed at developing a peptide-based drug delivery system, which could carry multiple cytotoxic units, increasing the efficacy of targeted drug delivery. We synthesized different NT4 adducts, where NT4 was covalently coupled to one, two, or three molecules of PTX. We also prepared a nanosystem in which a QD nanoparticle was linked to the tumor-targeting carrier, i.e., the NT4 peptide, and paclitaxel. The resulting NT4-QD-PTX nanodevice was designed to simultaneously detect and kill tumor cells.

## 2. Results

### 2.1. Paclitaxel for NT4 Multidrug Delivery

NT4 was solid-phase synthesized and conjugated with a variable number of paclitaxel units. Based on previous experience, we selected a Michael reaction between the thiol group of the cysteine residue, added to the c-terminal of NT4, and a maleimide group, reversibly linked to the C2-O′ oxygen of paclitaxel, to assemble the NT4 multidrug devices. Aiming to minimize any pharmacokinetic differences caused by the chemical nature of the linker(s), we designed a common spacer bringing a tunable (1–3) number of drug units, namely compounds **3a**–**c** reported in Scheme 1. Briefly, Paclitaxel (PTX) was regioselectively functionalized at the C-2′ OH by reaction with 6-azidohexanoic acid to provide derivative PTX-N_3_ (Scheme 1 step *a*). Then we prepared *O*-propargyl benzyl succinimides **2a**–**c** from commercially available hydroxy benzoic esters **1a**–**c**. In detail, classic propargylation of the phenol groups was followed by reduction of the ester group to the corresponding benzyl alcohols that were transformed into derivatives **2a**–**c** under Mitsunobu reaction conditions with maleimide, DIAD and Ph_3_P, and microwave irradiation (Scheme 1, steps *b-d*). Unexpectedly, the yield-limiting step of the procedure was insertion of the maleimide residue in benzyl position (i.e., step *d*). Several strategies were tested, and insertion was eventually achieved under Mitsunobu conditions from the corresponding benzyl alcohols. Working under microwave irradiation, we shortened the reaction time and scaled-up the procedure to isolate derivatives **2a**–**c** in reasonable amounts. A click copper(I)-catalyzed azide-alkyne cycloaddition (CuAAC) was used to anchor the drug(s) to PTX-N_3_. Carried out under classical conditions, CuSO_4_/sodium ascorbate in DMF/H_2_O, CuAAc led to the isolation of (multi)-functionalized systems, bearing 1 to 3 paclitaxel units **3a-c** in reasonable yields, as depicted in Scheme 1 step *e* (Appendix A). 

### 2.2. Conjugation of Mono, Bis, and Tris-PTX to NT4 

The NT4 tetrabranched peptide was prepared by automated solid phase synthesis, and after isolation was conjugated with a variable number of paclitaxel units (Figure 1). NT4 with a c-terminal cysteine moiety was left to react in DMF with paclitaxel carrying maleimide. The addition reaction was monitored by HPLC at different times: 1 h, 3 h, 20 h, and 48 h (not shown). When the starting materials were consumed, the reaction was stopped by simple removal of solvent under reduced pressure, followed by HPLC purification. Identity and purity of the final products **1**–**3** were confirmed by HPLC and mass spectrometry (not shown). 

### 2.3. Cytotoxicity Assay

Human colon adenocarcinoma HT-29 cells were treated with NT4-mono PTX, NT4-bis PTX, and NT4-tris PTX to evaluate the cytotoxicity (Figure 2). In order to minimize the effect of free PTX, released from the peptides by hydrolysis of the ester bond after as little as one hour [11], cells were incubated for 1 hour and then washed and incubated without the drug for 6 days.

The conjugation of more than one unit of drug with NT4 proved to have no impact on the overall cytotoxic effect, since the three compounds, NT4-mono PTX (**1**), NT4-bis PTX (**2**), and NT4-tris PTX (**3**) produced essentially the same effect in this experimental setting. The EC50s of NT4-mono PTX, NT4-bis PTX, and NT4-tris PTX were 1.9 × 10^−6^ M, 4.6 × 10^−6^ M, and 2.4 × 10^−6^ M, respectively. No cytotoxic effect was measured using unconjugated NT4 at the same concentrations (not shown).

### 2.4. Quantum Dot Decoration

Amine-modified QDs (Qdot 705 ITK amino PEG) were first conjugated with the bifunctional cross-linker sulfo-SMCC that provided the maleimide moiety for regiospecific conjugation with the Cys-derivative of NT4. PTX was conjugated with the amino residues of the QDs, after transformation into PTX-Nsuc (Scheme 2).

### 2.5. Paclitaxel for QDs

Paclitaxel bearing an activated carboxylic acid was prepared by reaction of PTX with succinic anhydride that occurs selectivity at C2′ OH to produce the derivative PTX-COOH (Scheme 3, step *a*). This derivative with its carboxylic acid residue was transformed into the corresponding *O*-succinimide derivative PTX-Nsuc with 51% yield (Scheme 3 step *b* and Appendix A). PTX-Nsuc carried the activated carboxylic group required for coupling PTX with the NH_2_ moieties presented on the QDs and a reasonably labile ester group that allowed the release of the cytotoxic moiety PTX.

QDs were first decorated with sulfo-SMCC that was subsequently used for coupling with NT4 in the last step of the reaction sequence (Scheme 2). A sub-stoichiometric quantity of sulfo-SMCC was used to ensure free amino groups on the nanoparticle for coupling with PTX. Activated PTX, PTX-Nsuc, was then used in excess to saturate all the amino groups left after the first reaction step. Finally, NT4 carrying a cysteine residue at the C-terminus was coupled orthogonally by a Michael reaction as already reported in Brunetti J. et al. [11] (Scheme 2). The estimated molar ratio of NT4:PTX, calculated on the amino groups of the QD yields a 1:4 ratio.

### 2.6. Cell Binding of NT4-QD-PTX

Binding and internalization experiments of NT4-QD on HT-29 human colon cancer cell line were analyzed by immunofluorescence, flow cytometry, and electron microscopy, as described in Brunetti J et al. [15]. Previous in vivo imaging experiments in mice bearing HT-29 xenografts showed much higher accumulation of NT4-QD than unconjugated QD in tumors [15].

Binding of NT4-QD-PTX to human colon adenocarcinoma cell line HT-29 was assessed by immunofluorescence and flow cytometry (Figure 3). Cells, incubated with scalar concentrations (20 µM, 10 µM and 5 µM) of NT4-QD-PTX or with unlabeled QD-PTX, showed a statistically significant dose-dependent fluorescent signal (Figure 3B).

HT-29 cells incubated with 20 nM NT4-QD-PTX showed a fluorescence signal two log higher than unlabeled QDs and the difference was statistically significant (Figure 3A).

Cell binding of NT4-QD-PTX was also analyzed in HT-29 cells by immunofluorescence (Figure 3C). NT4-QD-PTX bound HT-29 cells (red signal), while no signal was detected with unlabeled QD-PTX. Nuclei were stained with DAPI (blue signal).

### 2.7. Cytotoxicity Assays

Cytotoxicity of NT4-QD-PTX was tested on HT-29 cells and compared with the cytotoxicity of QD-PTX unlabeled with NT4. Different concentrations of QDs were incubated with HT-29 cells for one hour and then the cells were washed with medium and allowed to grow for 6 days (Figure 4). This procedure was essential to minimize the additive effect of free drug, released from the nanoparticles by hydrolysis of the ester bond. The EC50 of NT4-QD-PTX was 1.3 × 10^−8^ M and that of unlabeled QD-PTX was 4.5 × 10^−8^ M. 

## 3. Discussion

In the last years, molecularly targeted cancer therapies using antibodies, peptides, scFv, and related biomolecules are gaining momentum due to the possibility of improved drug potency and efficiency and minimal side effects. Peptides, in particular, can be used for their peculiar cytotoxicity or as cytotoxic drug carriers or as tumor specific probes to deliver nanosystems [17,18,19]. The development of selective tumor-targeting agents, whereby a single carrier delivers more than one unit of drug to the cancer tissue, would improve treatment efficacy and advance cancer therapy. We showed in previous studies that a tetrabranched peptide NT4 has extraordinary selectivity for different human cancers (e.g. colon, pancreas and bladder cancer) and can easily be coupled with chemical entities or nanoparticles to kill or image cancer cells [13,14,15,20,21]. Using in vivo bioluminescence imaging in an orthotopic mouse model of breast cancer, we previously found that conjugation of PTX with NT4 peptide led to increased therapeutic activity of the drug in vivo. NT4-PTX induced tumor regression, whereas treatment with unconjugated PTX only produced a reduction in tumor growth. Moreover, unlike PTX, NT4-PTX is very hydrophilic, which may improve its pharmacokinetic profile and allow less toxic dilution buffers to be used, thus decreasing the general toxicity of PTX [11]. NT4 derivatives are therefore very promising tumor-targeting agents that may enable high and broad cancer selectivity.

Considering the potential of PTX as a chemotherapeutic drug, and the extremely promising results obtained by us with NT4-PTX, PTX has been the drug of choice for the construction of new drug-armed chemotherapeutics that deliver multiple units of cytotoxic drug to tumor tissue. NT4 peptide was conjugated with one, two, and three units of paclitaxel and the cytotoxic effect was tested in vitro on the human colon adenocarcinoma cell line HT-29. The conjugation of more than one unit of paclitaxel with NT4 proved to have no additional impact on the overall cytotoxicity, indeed NT4-mono PTX, NT4-bis PTX and NT4-tris PTX had the same cytotoxic effect in vitro. 

Release of PTX from the ester bond by hydrolysis is considered crucial for the efficacy of peptide-bound drugs. In our previous study, the maleimido-propionic acid ester derivative NT4-PTX [11] achieved a higher EC50 than the three new NT4-PTX adducts reported above, which are hexanoic acid ester derivatives. We therefore infer that with these NT4-PTX adducts, increased loading of active drug on the carrier was not accompanied by efficient ester hydrolysis, resulting in poor efficacy. The lack of correlation between the number of PTX units per NT4 molecule and efficacy may also be ascribed to decreased solubility of the adducts. Every PTX moiety that is added to NT4 makes the peptide more hydrophobic and probably decreases its availability in the cell medium.

The second strategy we tested aimed at maximizing the drug delivery at the tumor site by developing NT4-based nanosystems, using NIR-emitting quantum dots. QD nanoparticles were linked to the tumor-targeted carrier, i.e., the NT4 peptide, and the cytotoxic moiety, PTX, to obtain a nanodevice NT4-QD-PTX designed to simultaneously detect and kill tumor cells. NT4-QDs, which we had already obtained and tested in previous in vitro and in vivo studies, were conjugated with PTX to obtain cancer-selective theranostics for in vivo diagnosis and therapy. Selective binding to HT-29 cells was confirmed by flow cytometry and confocal microscopy for NT4-QD-PTX with respect to unlabeled QD-PTX. NT4-QD-PTX cytotoxicity was tested in vitro on the HT-29 cell line and proved to be higher than that of QD-PTX. 

Since the advantages of tumor-selective uptake of NT4 peptides cannot be appreciated in experiments with a single cell population but are designed to be exploited in vivo, where increased local concentrations are evident and crucial, the results achieved so far with NT4-QD-PTX seem promising for further development as selective chemotherapeutics and specific in vivo tumor-cell tracers with diagnostic and therapeutic utility.

## 4. Materials and Methods 

### 4.1. Mono, Bis, and Tris-PTX Maleimide Construction

All the reactions were monitored by TLC on commercially available precoated silica gel 60 F 254 plates) and the products were visualized with acidic vanillin solution. Silica gel 60, 230–400 mesh, was used for column chromatography. ^1^H and ^13^C-NMR spectra were recorded with a Varian Gemini 200 or Varian Mercury Plus 400 using D-Chloroform (CDCl_3_) as solvent. The signal at 7.26 ppm of the residual CHCl_3_ in ^1^H-NMR spectra and at 77.0 ppm in the ^13^C-NMR spectra were used as reference. FT-IR spectra were recorded in CDCl_3_ solutions with a Perkin-Elmer 1600 FTIR Spectrometer (PerkinElmer, Waltham, MA, USA). ESI-MS spectra were measured either in positive and negative mode with a *J*EOL *J*MS700 MStation (JEOL Ltd. Akishima, Tokyo, Japan). Commercial reagents, catalysts, and ligands were used as obtained, unless otherwise stated, from freshly opened containers without further purification. Dry dichloromethane (DCM), diethyl ether (Et_2_O), tetrahydrofuran (THF), and dimethylformamide (DMF) were obtained from Pure Solv Micro devices (Sigma Aldrich, St. Louis, MO, USA Acetone was distilled over calcium chloride.

Synthesis of PTX-N_3_ (Scheme 1 step *a*)



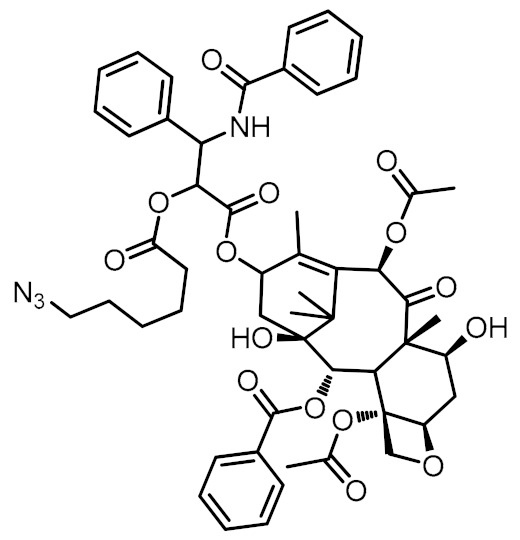



DMAP (6 mg, 0.05 mmol) was added to a solution of Paclitaxel (PTX, 200 mg, 0.23 mmol) in dry DCM (8 mL) under N_2_. The colorless solution was cooled to 0 °C, then 6-azido-hexanoic acid (44 mg, 0.28 mmol) was added, followed by DCC (58 mg, 0.28 mmol). The reaction mixture was stirred for 24 h, at reflux, under N_2_, then poured onto DCM (15 mL) and washed with a saturated solution of NH_4_Cl (3 × 40 mL) and a saturated solution of NaHCO_3_ (3 × 40 mL) followed by a saturated solution of NH_4_Cl (3 × 40 mL). The organic layer was dried over Na_2_SO_4_, filtered and evaporated. The crude product was purified by flash column chromatography using EP:AcOEt 1:1 as eluent to obtain PTX-N_3_ (195 mg, 85% yield). Spectroscopic data was in line with the literature [22]. ^1^H-NMR (400 MHz, CDCl_3_) δ 1.14 (s, 3H), 1.23 (s, 3H), 1.31–1.38 (m, 4H), 1.52–1.63 (m, 3H), 1.68 (s, 3H), 1.94 (s, 3H), 2.17 (s, 1H), 2.23 (s, 3H), 2.33–2.42 (m, 2H), 2.46 (s, 3H), 2.48–2.60 (m, 2H), 3.21 (t, *J* = 6.8 Hz, 2H), 3.82 (d, *J* = 7.2 Hz, 1H), 4.20 (d, *J* = 8.8 Hz, 1H), 4.31 (d, *J* = 8.8 Hz, 1H), 4.42–4.47 (m, 1H), 4.97 (d, *J* = 9.6 Hz, 1H), 5.51 (d, *J* = 3.6 Hz, 1H), 5.68 (d, *J* = 7.2 Hz, 1H), 5.96 (dd, *J* = 12.0 Hz, *J* = 2.8 Hz, 1H), 6.25 (t, *J* = 10.0 Hz, 1H), 6.29 (s, 1H), 6.88 (d, *J* = 9.2 Hz, 1H), 7.34–7.63 (m, 11H), 7.74 (d, *J* = 8.4 Hz, 2H), 8.13 (d, *J* = 8.4 Hz, 2H) ppm. FT-IR (CDCl_3_) N3 stretc. 2200 cm^−1^.

Synthesis of mono-, bis- and tris-propargyl succinimide derivatives **2a**–**c** (Scheme 1, steps *b–d*)

*b)* Propargylation. General procedure: 18-crown-6 (0.01 mmol) and propargyl bromide (6.00 mmol) were added in sequence under N_2_ to a solution of proper hydroxy-benzoic acid ethyl ester (**1a**–**3a**) in dry acetone and solid K_2_CO_3_ (6.00 mmol). The suspension was stirred at reflux for 24 h. Acetone was evaporated, water (30 mL) was added and the mixture was extracted with DCM (3 × 30 mL). The organic phase was dried over Na_2_SO_4_, filtered and evaporated to obtain target compounds in nearly quantitative yield, used without further purification.

4-prop-2-ynyloxy-benzoic acid ethyl ester (**1a**)



Following the general procedure, 600 mg (>98% yield) of target compound as a brown oil was obtained from 4-hydroxy-benzoic acid ethyl ester (**1a**, 500 mg, 3.01 mmol). ^1^H-NMR (200 MHz, CDCl_3_) δ 1.38 (t, *J* = 7.6 Hz, 3H), 2.55 (t, *J* = 2.4Hz, 1H), 4.35 (q, *J* = 7.6 Hz, 2H), 4.75 (d, *J* = 2.4Hz, 2H), 6.99 (m, 2H), 8.02 (m, 2H) ppm. FT-IR (CDCl_3_) 2258, 3308 cm^−1^.

3,4-Bis-prop-2-ynyloxy-benzoic acid ethyl ester (**1b**)



Following the general procedure, 297 mg (81% yield) of the target product was obtained from 3,4-dihydroxy-benzoic acid ethyl ester (**1b**, 258 mg, 1.42 mmol). ^1^H-NMR (400 MHz, CDCl_3_) δ 1.37 (t, *J* = 7.2 Hz, 3H), 2.53 (t, *J* = 2.4 Hz, 2H), 4.35 (q, *J* = 7.2 Hz, 2H), 4.80 (d, *J* = 2.4 Hz, 2H), 4.81 (d, *J* = 2.4 Hz, 2H), 7.06 (d, *J* = 9.2 Hz, 1H), 7.71-7.73 (m, 2H) ppm. ^13^C-NMR (100 MHz, CDCl_3_) δ 14.3, 56.6, 56.9, 60.9, 76.2, 76.4, 77.7, 77.9, 113.1, 115.4, 124.0, 124.2, 146.8, 151.2, 166.0 ppm. FT-IR (CDCl_3_) 2260, 3307 cm^-1^. ESI-MS: *m/z* = 208.08 [M + Na]^+^.

3,4,5-tris-prop-2-ynyloxy-benzoic acid ethyl ester (**1c**)



Following the general procedure, 830 mg (>98% yield) of target compound was obtained from ethyl gallate (**1c**, 500 mg, 2.52 mmol) [23]. ^1^H-NMR (400 MHz, CD_3_OD) δ 1.38 (t, *J* = 7.2 Hz, 3H), 2.86 (t, *J* = 2.4 Hz, 1H), 3.00 (t, *J* = 2.4 Hz, 2H), 4.36 (q, *J* = 7.2 Hz, 2H), 4.78 (d, *J* = 2.4 Hz, 2H), 4.83 (d, *J* = 2.4 Hz, 4H), 7.47 (s, 2H) ppm. FT-IR (CDCl_3_) 2256, 3308 cm^−1^.

*c)* Reduction. General procedure: A suspension with an excess of LiAlH_4_ (5.00 mmol) in dry THF was cooled to 0 °C and a 0.6 M solution of propynyloxy-benzoic esters in dry THF was added dropwise. The reaction mixtures were stirred overnight at r.t. under N_2_, then cooled to 0 °C and a saturated solution of NH_4_Cl was added. The mixtures were acidified to pH = 2 with HCl 10%, then extracted with DCM, dried over Na_2_SO_4_, filtered and evaporated to obtain the benzyl alcohols in nearly quantitative yields as pure compounds used without further purification.

Synthesis of (4-(prop-2-yn-1-yloxy)phenyl)methanol



Following the general procedure, the corresponding benzyl alcohol at 95% yield was obtained from 4-prop-2-ynyloxy-ethyl benzoate (369 mg, 1.81 mmol). ^1^H-NMR (200 MHz, CDCl_3_) δ 2.52 (t, *J* = 2.4 Hz, 1H), 4.64 (s, 1H), 4.70 (d, *J* = 2.4 Hz, 2H), 6.95–7.02 (m, 2H), 7.30–7.35 (m, 2H) ppm.

Synthesis of (3,4-bis(prop-2-yn-1-yloxy)phenyl)methanol



Following the general procedure, the corresponding benzyl alcohol was obtained at 86% yield from 3,4-*bis*-prop-2-ynyloxy-benzoic acid ethyl ester (140 mg, 0.54 mmol) [24].

Synthesis of (3,4,5-*tris*-prop-2-ynyloxy-phenyl)-methanol 



Following the general procedure, the corresponding benzyl alcohol was obtained at 95% yield from 3,4,5-*tris*-prop-2-ynyloxy-benzoic acid ethyl ester (1.40 g, 4.48 mmol) [22].

*d)* Mitsunobu reaction. General procedure: DIAD (1.1 mmol) was added dropwise to a solution of PPh_3_ (1.1 mmol) in dry THF (6 mL) cooled to 0 °C. After 5 minutes a solution of benzyl alcohol derivative (1.0 mmol) in dry THF (18 mL) was added, followed by maleimide (1.0 mmol) in dry THF (6 mL). The reaction was microwave heated as reported below. Then THF was evaporated and the crude products purified by flash column chromatography.

Synthesis of 1-(4-prop-2-ynyloxy-benzyl)-pyrrole-2,5-dione (**2a**)



Following the general procedure, after microwave irradiation at 100 °C for 1 h and 120 °C for 45 min, 4-prop-2-ynyloxy-phenyl)-methanol (153 mg, 0.94 mmol) was used to obtain a crude product which was purified by flash column chromatography using DCM as eluent to obtain **2a** (43 mg, 21% yield) as a colorless solid (m.p. 109–110 °C). ^1^H-NMR (400 MHz, CDCl_3_) δ 2.50 (t, *J* = 2.0 Hz, 1H), 4.61 (s, 2H), 4.66 (d, *J* = 2.0 Hz, 2H), 6.69 (s, 2H), 6.91 (d, *J* = 8.8 Hz, 2H), 7.30 (d, *J* = 8.8 Hz, 2H) ppm. ^13^C-NMR (100 MHz, CDCl_3_) δ 40.7, 55.7, 75.6, 78.4, 114.9, 129.3, 129.8, 134.1, 157.1, 170.4 ppm. FT-IR (CDCl_3_) 1711, 1732, 2258, 3308 cm^−1^.

Synthesis of 1-(3,4-*bis*-prop-2-ynyloxy-benzyl)-pyrrole-2,5-dione (**2b**)



Following the general procedure, after microwave irradiation at 120 °C for 80 min., 3,5-*bis*-prop-2-ynyloxy-phenyl)-methanol (70 mg, 0.32 mmol) was used to obtain a crude product that was purified by flash column chromatography using DCM:Et_2_O 10:1 as eluent to obtain **2b** (28 mg, 33% yield) as a colorless solid (m.p. 127–129 °C). ^1^H-NMR (400 MHz, CDCl_3_) δ 2.49 (t, *J* = 2.4 Hz, 1H), 2.51 (t, *J* = 2.4 Hz, 1H), 4.61 (s, 2H), 4.72 (d, *J* = 2.4 Hz, 2H), 4.73 (d, *J* = 2.4 Hz, 2H), 6.69 (s, 2H), 6.96–6.97 (m, 2H), 7.06 (d, *J* = 1.2 Hz, 1H) ppm. ^13^C-NMR (100 MHz, CDCl_3_) δ 41.0, 56.8, 56.9, 75.9, 76.0, 78.2, 78.4, 114.7, 115.3, 122.2, 130.1, 134.2, 147.1, 147.4, 170.3 ppm. FT-IR (CDCl_3_) 1711, 1732, 3307 cm^−1^. ESI-MS: *m/z* = 318.17 [M + Na]^+^.

Synthesis of 1-(3,4,5-*tris*-prop-2-ynyloxy-benzyl)-pyrrole-2,5-dione (**2c**)



Following the general procedure, after microwave irradiation at 80 °C for 30 min and 100 °C for 1 h, 3,4,5-*tris*-prop-2-ynyloxy-phenyl)methanol (100 mg, 0.37 mmol) was used to obtain a crude product which was purified by flash column chromatography using DCM:Et_2_O 30:1 as eluent to obtain **2c** (27 mg, 22% yield) as a colorless solid (m.p. 120–122 °C). ^1^H-NMR (400 MHz, CDCl_3_) δ 2.45 (t, *J* = 2.4 Hz, 1H), 2.51 (t, *J* = 2.4 Hz, 2H), 4.60 (s, 2H), 4.68 (d, *J* = 2.4 Hz, 2H), 4.73 (d, *J* = 2.4 Hz, 4H), 6.72 (s, 2H), 6.75 (s, 2H) ppm. ^13^C-NMR (100 MHz, CDCl_3_) δ 41.4, 57.0, 60.3, 75.2, 76.0, 78.2, 79.1, 108.8, 132.3, 134.2, 136.7, 151.5, 170.2 ppm. FT-IR (CDCl_3_) 1711, 3308 cm^−1^. ESI-MS: *m/z* = 372.17 [M + Na]^+^.

General Procedure for the Huisgen click reaction (Scheme 1, step *e*)

*e)* General procedure: CuSO_4_ (0.006 mmol) and sodium ascorbate (0.01 mmol) were added in sequence to a solution of maleimide-propynyloxy derivatives **2a**–**c** (0.04 mmol, in DMF 1mL). After 5 min, PTX-N_3_ (0.05 mmol in 1 mL DMF:H_2_O 3:1) was added and the mixture stirred at r.t. for 24–48 h. H_2_O (10 mL) was then added to the mixture and extracted with DCM (4 × 10 mL). The organic phases were washed with H_2_O until the complete removal of DMF, dried over Na_2_SO_4_, filtered and evaporated to dryness. The crude product was purified by flash column chromatography.

Synthesis of mono-PTX-maleimide adduct (**3a**) 



Following the general procedure from 1-(4-prop-2-ynyloxy-benzyl)-pyrrole-2,5-dione **2a** (10 mg, 0.04 mmol) and **PTX-N_3_** (49 mg, 0.05 mmol). The solution was stirred at r.t. for 24 h, and the crude product purified by flash column chromatography using AcOEt:EP 5:1 as eluent to obtain derivative **3a** (21 mg, 42% yield) as a white solid. ^1^H-NMR (400 MHz, CDCl_3_) δ 1.13 (s, 3H), 1.23 (s, 3H), 1.25–1.31 (m, 3H), 1.57–1.64 (m, 2H), 1.68 (s, 3H), 1.82–1.89 (m, 3H), 1.93 (s, 3H), 2.08–2.14 (m, 1H), 2.22 (s, 3H), 2.29–2.41 (m, 3H), 2.44 (s, 3H), 2.51–2.59 (m, 2H), 3.80 (d, *J* = 6.8 Hz, 1H), 4.18–4.32 (m, 4H), 4.41–4.47 (m, 1H), 4.60 (s, 2H), 4.96 (d, *J* = 8.8 Hz, 1H), 5.16 (s, 2H), 5.49 (d, *J* = 4.0 Hz, 1H), 5.68 (d, *J* = 7.2 Hz, 1H), 5.96 (dd, *J* = 9.6 Hz, *J* = 4.0 Hz, 1H), 6.22 (t, *J* = 8.0 Hz, 1H), 6.29 (s, 1H), 6.67 (s, 2H), 6.90 (d, *J* = 8.8 Hz, 2H), 7.07 (d, *J* = 9.2 Hz, 1H), 7.28–7.63 (m, 12H), 7.72 (d, *J* = 7.2 Hz, 2H), 8.13 (d, *J* = 7.2 Hz, 2H) ppm. ^13^C-NMR (100 MHz, CDCl_3_) δ 9.5, 14.7, 20.8, 22.1, 22.6, 23.8, 25.5, 26.7, 29.6, 33.3, 35.50, 35.52, 40.7, 43.1, 45.5, 49.8, 52.8, 58.4, 62.0, 71.7, 72.0, 74.0, 75.0, 75.5, 76.4, 79.0, 81.0, 84.4, 114.8, 122.5, 126.5, 127.0, 128.4, 128.6, 129.04, 129.08, 129.1, 129.8, 130.1, 131.9, 132.7, 133.64, 133.67, 134.1, 136.8, 142.6, 144.1, 157.7, 166.9, 167.1, 168.1, 169.8, 170.4, 171.1, 172.2, 203.7 ppm. ESI-MS: *m/z* = 1256.42 [M + Na]^+^ (Appendix A).

Synthesis of bis-PTX-maleimide adduct (**3b**)



Following the general procedure from 1-(3,4-*bis*-prop-2-ynyloxy-benzyl)-pyrrole-2,5-dione **2b** (15 mg, 0.05 mmol) and **PTX-N_3_** (49 mg, 0.05 mmol). The solution was stirred at r.t. for 24 h, and the crude product purified by flash column chromatography using AcOEt:EP 5:1 as eluent to obtain derivative **3b** (27 mg, 23% yield) as a pale brown solid. ^1^H-NMR (400 MHz, CDCl_3_) δ 1.12–1.25 (m, 18H), 1.56–1.75 (m, 10H), 1.75–1.92 (m, 16H), 2.04–2.10 (m, 4H), 2.20 (s, 6H), 2.26–2.37 (m, 6H), 2.43 (s, 6H), 2.49–2.62 (m, 4H), 3.79 (d, *J* = 7.2 Hz, 2H), 4.18 (d, *J* = 7.6 Hz, 2H), 4.29 (d, *J* = 8.4 Hz, 2H), 4.41–4.46 (m, 2H), 4.55 (s, 2H), 4.95 (d, *J* = 9.6 Hz, 2H), 5.17 (s, 1H), 5.19 (s, 1H), 5.50 (d, *J* = 4.0 Hz, 2H), 5.66 (d, *J* = 6.8 Hz, 2H), 5.94 (dd, *J* = 4.0 Hz, *J* = 9.2 Hz, 2H), 6.20 (t, *J* = 8.0 Hz, 2H), 6.29 (s, 2H), 6.66 (s, 2H), 6.93–6.95 (m, 1H), 7.15–7.18 (m, 2H), 7.29–7.63 (m, 24H), 7.71 (d, *J* = 7.6 Hz, 4H), 8.12 (d, *J* = 7.2 Hz, 4H) ppm. ^13^C-NMR (100 MHz, CDCl_3_) δ 9.6, 14.8, 20.8, 22.1, 22.7, 23.8, 25.5, 26.8, 29.6, 33.3, 35.4, 35.6, 43.1, 45.6, 49.8, 53.0, 58.4, 71.7, 72.0, 74.1, 75.0, 75.6, 79.1, 81.0, 84.4, 126.7, 127.1, 128.5, 128.6, 128.7, 129.0, 129.2, 130.2, 131.9, 132.8, 133.6, 133.7, 134.2, 137.0, 142.7, 167.0, 167.1, 168.2, 169.8, 170.4, 171.1, 172.3, 203.7 ppm. FT-IR (CDCl_3_) 1713, 2248, 2258, 2944, 3447, 3524, 3567 cm^-1^. ESI-MS: *m/z* = 2303.88 [M + Na]^+^ (Appendix A).

Synthesis of tris-PTX-maleimide adduct (**3c**)



Following the general procedure from 1-(3,4,5-*tris*-prop-2-ynyloxy-benzyl)-pyrrole-2,5-dione **2c** (8 mg, 0.02 mmol) and **PTX-N_3_** (49 mg, 0.05 mmol). The solution was stirred at r.t. for 48 h, and the crude product was purified by flash column chromatography using DCM:MeOH 10:1 as the eluent to obtain derivative **3c** (12 mg, 16% yield) as a white solid. ^1^H-NMR (400 MHz, CDCl_3_) δ 1.12 (s, 9H), 1.25 (s, 9H), 1.25–1.40 (m, 6H), 1.54–1.62 (m, 6H), 1.66–1.96 (m, 12H), 1.79–1.87 (m, 4H), 1.91 (s, 9H), 2.05–2.11 (m, 3H), 2.20 (s, 9H), 2.25–2.36 (m, 9H), 2.42 (s, 9H), 2.49–2.66 (m, 6H), 3.42–3.50 (m, 3H), 3.78 (d, *J* = 7.2 Hz, 3H), 4.09–4.31 (m, 15H), 4.40–4.53 (m, 3H), 4.95 (d, *J* = 9.6 Hz, 3H), 5.07–5.15 (m, 4H), 5.48 (d, *J* = 4.0 Hz, 3H), 5.66 (d, *J* = 6.8 Hz, 3H), 5.93 (dd, *J* = 6.8 Hz, *J* = 4.4 Hz, 3H), 6.17–6.23 (m, 3H), 6.29 (s, 3H), 6.66 (s, 2H), 6.69 (s, 2H), 7.31–7.53 (m, 27H), 7.59–7.73 (m, 12H), 8.12 (d, *J* = 8.4 Hz, 6H) ppm. ^13^C (100 MHz, CDCl_3_) δ 9.6, 14.8, 20.8, 22.05, 22.08, 22.7, 23.8, 24.9, 25.5, 26.7, 29.60, 29.64, 29.68, 33.2, 33.9, 43.1, 49.1, 49.8, 58.4, 75.5, 79.0, 80.9, 84.3, 84.4, 126.7, 127.1, 128.64, 128.69, 129.0, 129.1, 130.1, 133.6, 136.9, 166.9, 167.1, 168.2, 169.8, 170.3, 171.1, 172.3, 203.7. FT-IR (CDCl_3_) 1713, 2801, 2860, 3308, 3435 cm^−1^. ESI-MS: *m/z* = 1665.17 [(M + 2H)/2]^+^ (Appendix A).

### 4.2. Peptide Synthesis 

Solid-phase synthesis was carried out on a MultiSynTech Syro automated multiple peptide synthesizer (Witten, Germany), employing Fmoc chemistry with 2-(1H-benzotriazole-1-yl)-1,1,3,3-tetramethyluronium hexafluorophosphate/*N*,*N*-diisopropylethylamine (HBTU/DIPEA) activation. NT4-Cys was synthesized on TG S RAM resin using Fmoc-Cys(Trt)-OH as first, and Fmoc-PEG12-OH (Iris Biotech, Germany) as the second coupling step. Then two coupling steps with Fmoc-Lys(Fmoc)-OH were used to build the core. Pyro-glu-*O*-pentachlorophenylester was used for the N-terminal acid of the neurotensin sequence. Peptides were cleaved from the resins and deprotected by treatment with trifluoroacetic acid containing water and triisopropylsilane (95:2.5:2.5) for 1.5 hours at room temperature. After precipitation with diethyl ether, branched peptides were purified by RP-HPLC. Final peptide purity was confirmed to exceed by 99% C18 Jupiter column HPLC (Phenomenex, 300 Å, 5 μm, 250 × 4.6 mm) using 0.1% TFA/water as eluent A and acetonitrile as eluent B with a linear gradient from 75% A to 65% A in 40 min. All peptides were characterized by UltraflexIII MALDI TOF/TOF mass spectrometry (Bruker Daltonics, Bremen, Germany). Commercial reagents, catalysts, and ligands were used without further purification from freshly opened containers, unless otherwise stated. 

### 4.3. NT4-mono PTX, NT4-bis PTX, and NT4-tris PTX construction

Mono, bis or tris PTX maleimide derivatives reacted with NT4-Cys, in DMF, at maleimide:peptide:DIPEA molar ratio 5:1:1. The reaction mixture was left under stirring for 20 hours or 48 hours (for NT4-tris PTX). Then each conjugate was purified by RP-HPLC on a semi-preparative C18 Jupiter column (Phenomenex, 300 Å, 10 μm, 250 × 10 mm) using 0.1% TFA/water as eluent A and acetonitrile as eluent B with a two-step linear gradient from 80% A to 20% A in 30 min and from 20% A to 5% A in 5 min. All conjugates were characterized by UltraflexIII MALDI TOF/TOF mass spectrometry (Bruker Daltonics, Bremen, Germany). The final products were more than 90% pure, calculated on HPLC spectra. MS spectra showed the main peaks at the expected molecular masses: 8958 [NT4-mono PTX + H]^+^, 10005 [NT4-bis PTX + H]^+^ and 11052 [NT4-tris PTX + H]^+^.

Synthesis of PTX-COOH (step *a* of Scheme 3)



Succinic anhydride (0.014 g, 0.14 mmol) and DMAP (0.003 g, 0.02 mmol) were added in sequence to a solution of paclitaxel (**PTX**, 0.1 g, 0.12 mmol) in 4 mL dry pyridine. The solution was stirred for 3 h at r.t under N_2_. The solvent was vacuum-evaporated to obtain a colorless oil. The crude product was dissolved in 20 mL DCM and washed with a saturated solution of NH_4_Cl (3 × 40 mL), followed by water (3 x 40 mL), to remove residual pyridine. The organic layer was dried over Na_2_SO_4_, filtered and evaporated to obtain PTX-COOH as a white solid (43 mg, 38% yield). Spectroscopic data was in line with the literature [25]. ESI-MS spectra were measured either in the negative or positive mode: *m/z* = 852.17 [M − succ]^−^, 952.17 [M − H]^−^, 1906.00 [2M − H]^−^; 976.17 [M + Na]^+^**.**

Synthesis of PTX-NSuc (step *b* of Scheme 3)



Diphenyl chlorophosphate (268.6 mg, 1.0 mmol) and triethylamine (101.19 mg, 1.0 mmol) were added in sequence to a solution of N-hydroxysuccinimide (115 mg, 1.0 mmol) in dry DCM (4 mL) and the mixture stirred at r.t. for 2 hours [26]. The solvent was evaporated and the crude product was dissolved in ethyl acetate (10 mL) and washed with a saturated aqueous solution of NaCl (3 × 10 mL). The organic layer was dried over Na_2_SO_4_, filtered and evaporated to obtain N-succinimidyl diphenylphosphate (SDDP) in nearly quantitative yield, which was used without purification in the next step. Freshly prepared succinimidyl diphenylphosphate (18 mg, 0.053 mmol) and triethylamine (14 mg, 0.14 mmol) were added in sequence to a solution of PTX-COOH (40 mg, 0.035 mmol) in dry acetonitrile (1.2 mL). The mixture was stirred under nitrogen for 6 hours, and the solvent vacuum-evaporated. The crude product was purified by flash column chromatography to obtain PTX-Nsuc (18 mg, 51% yield) as a white solid. Spectroscopic data was in line with previous reports [23]. ^1^H-NMR (300 MHz, CDCl3) δ 8.14 (d, *J* = 6.9, 2H), 7.71 (d, *J* = 6.9, 2H), 7.61 (t, *J* = 7.5 Hz,1H), 7.54–7.33 (m, 10H), 6.30 (s, 1H), 6.28 (t, *J* = 8.4, 1H), 5.98 (dd, *J* = 9.3, 3.0, 1H), 5.69 (d, *J* = 7.5, 1H), 5.48 (d, *J* = 3.0, 1H), 4.99 (d, *J* = 8.4, 1H), 4.45 (dd, *J* = 10.7, 6.2 1H), 4.32 (d, *J* = 7.8, 1H), 4.21 (d, *J* = 8.7, 1H), 3.82 (d, *J* = 7.2, 1H), 2.91–2.51 (m, 8H), 2.49 (s, 3H), 2.38 (t, *J* = 7.5 Hz, 2H), 2.23 (s, 3H), 2.21–2.02 (m, 3H), 1.95 (s, 3H), 1.86 (m, 1H), 1.68(s, 3H), 1.24 (s, 3H), 1.14 (s, 3H) ppm. ESI-MS: *m/z* = 1073.25 [M + Na]^+^ (Appendix A).

### 4.4. NT4-QD-PTX Construction

Qdot 705 ITK amino PEG (Qdot 705 ITK™ Amino (PEG) Quantum Dots, Molecular Probes) were transferred to phosphate buffered saline (PBS, pH 7.4) using a PES 100K MWCO protein concentrator (Thermo Fisher Scientific, Waltham, MA, USA) spun at 3800× *g* at 4 °C. Quantum dots were then conjugated with sulfo-SMCC (sulfosuccinimidyl-4-(*N*-maleimidomethyl)cyclohexane-1-carboxylate; Thermo Fisher Scientific) for 1 hour at room temperature on a rotator ALC 120R (Thermo Fisher Scientific) to generate a maleimide-activated surface on the QDs, using a QDs:sulfo-SMCC 1:20 molar ratio, which corresponds to a ratio of NH_2_:sulfo-SMCC 5:1. Then PTX-Nsuc (50 equivalents with respect to QDs) was added to the mixture and left to react for 1 hour. Free sulfo-SMCC and free PTX-Nsuc were removed using a NAP-5 column (GE Healthcare, Chicago, IL, USA). The isolated maleimide-QD-PTX was incubated with NT4-Cys for 1 hour at room temperature on the rotator, using a QDs:NT4-Cys 1:50 molar ratio. Total of 10 mM 2-mercaptoethanol was added to the reaction and left for an additional 30 minutes to cap unreacted maleimides. NT4-conjugated QD-PTX was further purified on a Superdex 200 column, eluting with 20 mM phosphate buffer (pH 7.4). The concentration of quantum dots was examined using a spectrophotometer at 532 nm and an extinction coefficient of 2.1 × 10^6^ (mol/L)^−1^ cm^−1^. The ^1^H-NMR spectrum showed the signals of PTX and NT4.

### 4.5. NMR Measurements

Solution NMR spectra were recorded at 298 K on Bruker AVANCEIII-HD (Bruker, Billerica, MA, USA) and AVANCE NEO NMR spectrometers operating at 950 MHz and 900 MHz (^1^H Larmor frequency), respectively. Both instruments were equipped with triple resonance cryo-probes for high sensitivity. The samples were prepared by dissolving QDs in PBS with 10% D_2_O. The spectra for NT4-QD-PTX, QD-PTX, and undecorated QDs were acquired with 8192, 16384, and 1024 scans, respectively. All the spectra were recorded with a relaxation delay of 3 s.

### 4.6. Cell Cultures

Human colon adenocarcinoma HT-29 cells were grown in McCoy’s 5a Medium supplemented with 10% fetal bovine serum, 200 µg/mL glutamine, 100 µg/mL streptomycin, and 60 µg/mL penicillin. They were maintained under 5% CO_2_. The cell line was purchased from ATCC (The Global Bioresource Center, Manassas, VA, USA).

### 4.7. In Vitro Cytotoxicity Assay

HT-29 cells were plated at a density of 5 × 10^3^ per well in 96-well microplates. Different peptide concentrations (NT4-mono PTX, NT4-bis PTX, NT4-tris PTX, NT4-QD-PTX and undecorated QD-PTX) were added 24 h after plating. Cells were washed after 1 h incubation and then left for 6 days at 37 °C in the same medium. Growth inhibition was assessed by 3-(4,5-dimethylthiazol-2-yl)-2,5-diphenyltetrazolium bromide (MTT). The experiment was performed twice in triplicate. EC50 values were calculated by non-linear regression analysis using GraphPad Prism 5.03 software (San Diego, CA, USA).

### 4.8. NT4-QD-PTX Binding by Flow Cytometry

100,000 cells/experiment were incubated in 96-well U-bottom plates for 30 minutes at room temperature with different concentrations of NT4-QDs (5 nM, 10 nM, and 20 nM) in PBS-EDTA 5 mM-BSA 0.5%. Flow cytometric analysis was performed with 10,000 events using a BD FACS Canto II instrument (BD, NJ. USA). Assays were performed in triplicate and the flow cytometry results were analyzed by nonlinear regression analysis using GraphPad Prism 5.03 software. 

### 4.9. NT4-QD-PTX Binding by Immunofluorescence

The binding of NT4-QD-PTX and unlabeled QD-PTX was tested in HT-29 cells. Total of 3 × 10^4^ cells/well were seeded on 24-well plates, grown for 24 hours and then incubated with 20 nM NT4-QDs or with unlabeled QDs for 30 min at 37 °C in PBS-1% BSA. After 30 minutes of incubation, the cells were fixed with PBS 4% formalin. Each step was followed by three washes in PBS. Samples were mounted using Prolong Gold antifade with DAPI (Molecular Probes, Eugene, OR, USA) and analyzed by confocal laser microscope (Leica TCS SP5) with 380 nm 𝜆ex and 680–750 nm 𝜆em and 380 nm 𝜆ex and 450–470 nm 𝜆em for NIR-QDs and DAPI, respectively. All images were processed using ImageJ software (NIH, Rockville, MD, USA).

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
