# Peer review of "A New NT4 Peptide-Based Drug Delivery System for Cancer Treatment"

_molecules, 2020, doi:10.3390/molecules25051088_

Round 1
Reviewer 1 Report
The authors present a well-constructed study on the development of a targeted system for the delivery of multiple copies of a chemotherapeutic agent to tumour cells.
This work builds upon previous studies by the authors with a specific branched targeting peptide, NT4, and the background and motivation for the current work is very clearly set out. The synthesis of novel ligatable scaffolds for the selective attachment of 1, 2, or 3 paclitaxel units by CuAAC chemistry is presented, along with the subsequent conjugation of mono, bis, and tris PTX to NT4. The chemistry is generally well described, although I should like to see all relevant yields quoted in the schemes. All compounds are characterised appropriately, but I should like however, to see IR data for the key azido (PTX-N3) and propargylated derivatives (1a-c) included. For the final conjugates 1, 2 and 3, the authors should also include the HPLC data confirming purity in the Supplementary Information.
Cytotoxicity studies with the 1-3 are adequately described, and a reasonable rationalisation is provided for the unexpected lack of toxicity upon increasing the number of PTX copies. The authors also present some promising results for the tumour-selective uptake and cytotoxicity of NT4-targeted PTX-coated quantum dots. These novel materials appear to have some promise as theranostics - it would be good if the authors could comment on the number of copies of PTX that are coupled per nanoparticle in this case, and the scope to vary levels of drug and targeting unit in future studies.
Overall, this is an original and well-conceived piece of work that should be of interest to a range of researchers with interests in organic chemistry and targeted drug delivery. I am happy to recommend publication subject to the points above being addressed and the following additional minor issues:
Page 6, line 132: "hanging from", change to "presented on"
Page 6, line 137: "Sulfo-SMCC was used short", should be "A sub-stoichiometric quantity of Sulfo-SMCC was used..."??
Page 6, line 143: "was analysed", should be "were analysed"
Page 9, line 225: indicate whetehr +ve or -ve mode was used
Page 10-11: Compounds 1a-c appear to be un-numbered.
Page 11, line 289: OH group in wrong orientation
Page 12: N's in wrong orientation on structures
Page 13: Structure of 3a: OH and carbonyl overlap
Page 16, line 419: Present data as [M+H]+ etc
Page 16, line 430: Is this a combination of +ve and -ve mode data? Should be presented as 2 separate experiments.
Page 19, line 507: Superscripts for NMR
Reviewer 2 Report
The manuscript written by Bracci et al is a manuscript for NT4, a tetra-branched peptide-based drug delivery system for cancer treatment.
The authors start from the brief introduction of drug delivery systems, peptide-based systems used in drug delivery, drawbacks and various peptide-based drug delivery nanosystems.
The authors also mention and explain their development of peptide-based drug delivery system using the tetra-branched peptide NT4 in a human colon adenocarcinoma cell line HT-29. Following this, they mention the results, discussion, materials and methods.
The manuscript is written and the results mention that NT4-QD-PTX seems promising for further development as selective chemotherapeutics, and is suitable for publishing in "Molecules" after minor revision.
I suggest the authors may want to add a little bit more information about how their NT4 peptide-based drug delivery system is better than other similar examples out there with specific references and examples.
